# Identification of Visible Lines in Pm-like W$^{13+}$

**Priti** [1,*] , **Kota Inadome** [2], **Mayuko Funabashi** [2], **Nobuyuki Nakamura** [1,2,†] , **Hiroyuki A. Sakaue** [1], **Izumi Murakami** [1,3] **and Daiji Kato** [1,4]

1   National Institute for Fusion Science, National Institutes of Natural Sciences, Toki 509-5292, Gifu, Japan
2   Institute for Laser Science, The University of Electro-Communications, Chofu 182-8585, Tokyo, Japan
3   Department of Fusion Science, The Graduate University for Advanced Studies, SOKENDAI, Toki 509-5292, Gifu, Japan
4   Interdisciplinary Graduate School of Engineering Sciences, Kyushu University, Kasuga 816-8580, Fukuoka, Japan
*   Correspondence: priti.priti@nifs.ac.jp
†   Affiliation 2 is the author's primary organization.

**Abstract:** To provide spectroscopic data for W$^{13+}$, the present work is focused on the analysis of spectra observed in the visible range, using a compact electron beam ion trap (CoBIT). Line identification is done by using a collisional radiative model, along with sophisticated structure calculations from FAC and GRASP2018. Most of the identified lines belong to magnetic dipole (M1) transitions between the levels of the $4f^{12}5p^1$ and $4f^{13}$ configurations.

**Keywords:** visible spectra; compact electron beam ion trap; collisional radiative model; atomic structure

## 1. Introduction

Tungsten is planned to be used as a plasma-facing material in the ITER and is expected to be one of the main impurities. Since the temperature of the diverter region of ITER is expected to be a few hundred eV, a large fraction of the plasma is anticipated to contain tungsten ions in charge states mainly from three to fifteen times ionized, W$^{3+}$–W$^{15+}$ [1–3]. Therefore, spectroscopy of a few times ionized tungsten, from lower charge states, is highly relevant to diverter plasmas. Moreover, the visible lines are particularly useful for fusion plasma diagnostics, because optical fibers are available, to avoid direct neutron irradiation of the detectors. However, there is still a shortage of spectroscopic data for the lower charge states of W ions (W$^{7+}$–W$^{26+}$), which calls for more spectral measurements and accurate line identifications. In this study, we are interested in the visible spectra of Pm-like W$^{13+}$ ions. It has $4f$ open shells and is situated near the level crossing, i.e., the orbital energies of the $4f$, $5s$, and $5p$ subshells are very close [4]. Consequently, obtaining reliable and accurate atomic data for this Pm-like sequence is challenging.

Several previous studies have examined the ion spectra of Pm-like W$^{13+}$ in the visible region, and the $5p \rightarrow 5s$ spectrum in the EUV region, both experimentally and theoretically [5]. Here, we focus on reports available for visible transitions. Kramida and Shirai [2] predicted that the ground state of W$^{13+}$ is from the $5s^24f^{13}$ configuration viz. $(4f^{13})_{J=7/2}$, and the first excited state $(4f^{13})_{J=5/2}$ is situated at 18,000 cm$^{-1}$ from the ground state. A few lower energy levels, calculated by using flexible atomic code (FAC), have also been reported recently by Singh et al. [6]. Compact EBITs, in Shanghai [7,8] and Tokyo [9], were used in the attempt to find the visible transition between the fine structure splitting of the ground state. In the study with a compact EBIT in Tokyo (CoBIT), several visible lines were assigned to W$^{13+}$, but only the transition from the ground state splitting was calculated extensively, and other lines were unidentified. Very recently, Liu et al. [10] made an effort to identify the lines, and reported the visible transitions and theoretical spectra for W$^{13+}$ ions.

In this work, we identified the visible transitions of W$^{13+}$ ions observed at CoBIT [9] using detailed collisional-radiative (C-R) model calculations, along with configurational interaction (CI) structure calculations, using FAC. To aid the identification's reliability, we

have also performed the structure calculation, within the multi-configuration Dirac–Fock (MCDF)+CI method, using GRASP2018 [11].

## 2. Theoretical Model

The present C-R model includes thousands of excited levels originating from the $5s^2 4f^{13}$, $5s^1 4f^{14}$, $5s^2 4f^{12} 5p^1$ configuration, and single, as well as some important double, excitation from this configuration to higher orbitals, $viz.$ $5s^2 4f^{11} 5p^2$, $5s^2 4f^{12} 5l^1 (l = d, f)$, $5s^1 4f^{13} 5l^1 (l = p, d, f)$, $4f^{14} 5l^1 (l = p, d, f)$, $4f^{13} 5p^2$, $4f^{13} 5p^1 5d^1$, $5s^1 4f^{12} 5p^2$, $5s^1 4f^{12} 5p^1 5d^1$, and $5s^2 4f^{11} 5p^1 5d^1$. To improve the accuracy of the energy levels, various correlations, by considering the configurations from double excitation to 5(d,f), excitation to 6(s,p,d,f) orbitals, and excitation from 4d orbitals $viz.$ $4f^{13} 5p^1 5f^1$, $4f^{13} 5ll'^1 (l, l' = d, f)$, $5s^1 4f^{12} 5p^1 5f^1$, $5s^1 4f^{12} 5ll'^1 (l, l' = d, f)$, $5s^2 4f^{11} 5p^1 5f^1$, $5s^2 4f^{11} 5ll'^1 (l, l' = d, f)$, and $5s^2 4f^{12} 6l^1 (l = s, p, d, f)$, $5s^1 4f^{13} 6l^1 (l = s, p, d, f)$, $4f^{14} 6l^1 (l = s, p, d, f)$, $5s^2 4f^{11} 5p^1 6l^1 (l = s, p, d, f)$, $5s^1 4f^{12} 5p^1 6l^1 (l = s, p, d, f)$, $4f^{13} 5p^1 6l^1 (l = s, p, d, f)$, and $4d^9 5s^2 4f^{14}$, $4d^9 5s^2 4f^{13} 5p^1$, $4d^9 5s^2 4f^{12} 5p^2$, $4d^9 5s^1 4f^{14} 5p^1$, respectively, are also added, only in the structure calculation.

The populations of the excited levels are obtained by solving the rate balance equations for all excited states simultaneously. The present model primarily considered population or depopulation by electron impact excitation (de-excitation), ionization and radiative decay, and radiative recombination. In the steady state, the rate balance equation for an excited state $j$, with the normalization condition $\sum_j n_j = 1$, is simply given by:

$$\sum_{\substack{i \\ i \neq j}} k_{ij} n_i n_e + \sum_{i>j} A_{ij} n_i + n_+ n_e k_{+j} - \sum_{\substack{i \\ i \neq j}} k_{ji} n_j n_e - \sum_{i<j} A_{ji} n_j - n_j n_e k_{j+} = 0.$$

here, $A_{ij}$, $n_j$, $n_+$, and $n_e$ are the transition probability from level $i \rightarrow j$, the population of the $j$th level, the population of the ionic state, and the electron density, respectively. $k_{ij}$, $k_{j+}$, and $k_{+j}$, represent the rate of electron impact excitation (de-excitation), the rate of electron impact ionization, and the rate of radiative recombination, respectively. Since the EBIT plasma has a quasi-monoenergetic electron distribution, all the rates were calculated using a narrow Gaussian electron energy distribution function, using the FAC [12]. In total, more than a million cross-sections among all the states, and millions of radiative decay channels (E1, E2, E3, M1, M2, and M3), are accounted for in the model.

To ensure line identification from the C-R model, transition energies and transition probabilities are also calculated from the atomic state wave functions, obtained from MCDF calculation combined with the RCI approach, using GRASP2018 [11]. In the MCDF approximation, the wave function for an atomic state, approximated by an atomic state function (ASF), is expressed as a linear combination of configuration state functions (CSFs) of the same angular momentum and parity. To obtain the final ASFs, we start with a wave function calculation using a self-consistent-field (SCF) procedure, based on the Dirac–Coulomb Hamiltonian, and various correlation effects are included by enlarging the basis set through systematically increasing CSFs layer by layer, using an active set approach [13]. We will only present here the model space, for the details of the theory see the references [11,14].

In the present calculation, all the electrons are divided into two parts; electrons in the $4f$ and $5p$ orbitals are taken as valence electrons, and in other inner orbitals as core electrons. Correspondingly, the correlation is taken as the interaction between the valence electrons and valence electrons with core electrons. Valence–valence (VV), and core–valence (CV) are included via CSFs generated from restricted single–double (SD) excitation, i.e., SD excitation from $4s$, $4p$, $4d$, $4f$, $5s$, and $5p$, with the restriction that only one electron can excite from $4s$, $4p$, and $4d$ orbitals at a time, to $5p, 5d, 5f, 5g$, and SD-excitation only from $4f$, $5s$ and $5p$ up to $\{7s, 7p, 7d, 7f\}$. Energies for $4f^{13}$ and $4f^{12} 5p^1$ configurations are optimized for J = 1/2 to J = 15/2 states. The active set space is defined as follows;

| W$^{13+}$ | CSF's |
|---|---|
| DF = $\{4s^2 4p^6 4d^{10} 5s^2 4f^{13}\}$, $\{4s^2 4p^6 4d^{10} 5s^2 4f^{12} 5p^1\}$ | 70 |
| AS1 = DF + $\{5p, 5d, 5f, 5g\}$ | 2,565,429 |
| AS2 = AS1 + $\{6s, 6p, 6d, 6f\}$ | 3,082,728 |
| AS2 = AS1 + $\{7s, 7p, 7d, 7f\}$ | 3,994,266 |

While optimizing the outer layer, only the new ones are optimized and all the inner layers are kept fixed in the variational procedure. Further, relativistic configuration interaction (RCI) calculations are performed in order to include the Breit interaction, i.e., transverse photon interaction in the low-frequency limit and leading QED effects from self-energy correction (SE) and vacuum polarization (VP). We have calculated the level energies and transition probability for M1 transitions in W$^{13+}$.

### 3. Results and Discussion

Figure 1 shows the experimental spectra in the visible regions, observed using CoBIT at the University of Electro-Communications. In summary, multiply charged tungsten ions were produced in CoBIT, which comprises an electron gun, a drift tube, an electron collector, and a high-critical-temperature superconducting magnet. The drift tube consists of three cylindrical electrodes (DT1, DT2, and DT3), that function as an ion trap, by applying a positive potential ($\sim$30 V) at both ends relative to the middle electrode. The electron gun emits an electron beam that is compressed by the axial magnetic field (typically 0.08 T) generated by the magnet surrounding the drift tube, and subsequently accelerated towards the drift tube. The highly compressed electron beam forms a trap potential in the radial direction, and sequentially ionizes the ions trapped in the drift tube. The electron beam energy in CoBIT is primarily determined by the voltage difference between the cathode and DT2, but it may experience a systematic shift due to the electric field leakage and negative space charge shift. Nevertheless, the total estimated systematic shift is expected to be less than $-10$ eV [15]. Tungsten was continuously introduced into the trap through a gas injector, as a vapor of W(CO)$_6$, and the injection flow rate was meticulously regulated, using a variable leak valve to maintain a good charge state distribution. Visible spectra were observed using a commercial Czerny–Turner type of spectrometer, with a 1200 mm$^{-1}$ grating, blazed at 400 nm, for wavelength determination. The wavelength calibration was performed using emission lines from several standard lamps placed outside CoBIT, and the uncertainty in the wavelength calibration was estimated from reproducibility to be about $\pm$0.05 nm. Details of the experiment can be found in previous references [9,16].

Spectra were observed by varying the electron beam energy between 220 eV and 280 eV and tracking the energy dependence of the spectra, different charge state lines are assigned. At lower energies (220 eV), the W$^{11+}$ lines are dominant, however, as the electron energy increases to 240 eV, the W$^{12+}$ lines become dominant. W$^{11+}$ and W$^{12+}$ lines have already been identified [17,18]. As the energy was further increased up to 280 eV (which is greater than the ionization energy of W$^{12+}$, i.e., 258 eV), the lines emitted from W$^{13+}$ become dominant. The observed spectrum is shown in Figure 1, which indicates that some lines, identified as higher charge state lines, start appearing at lower electron beam energies. This may be attributed to ionization from metastable states, present near 5–20 eV for W$^{12+}$ ions and 10–15 eV for W$^{13+}$ ions.

Further, our theoretically simulated spectra from the C-R model, at 280 eV (the same energy at which W$^{13+}$ lines are observed in the experimental spectra) and at electron densities of $1 \times 10^{10}$ cm$^{-3}$ and $5 \times 10^{10}$ cm$^{-3}$, respectively, along with the transition probabilities, are presented in Figure 2. As an EBIT generates highly charged ions by successively ionizing trapped ions with a quasi-monoenergetic electron beam, a theoretical spectrum is obtained by including the rates of various processes, calculated using a narrow Gaussian electron energy distribution function with 5eV FWHM. On comparison of the line intensities of different lines in the theoretical and experimental spectra, a few dominant lines from W$^{13+}$ are identified. In the theoretical spectra, a few additional lines, other than

those observed in the experiment, are observed. This might be due to the very low intensity of these lines in the experimental spectra.

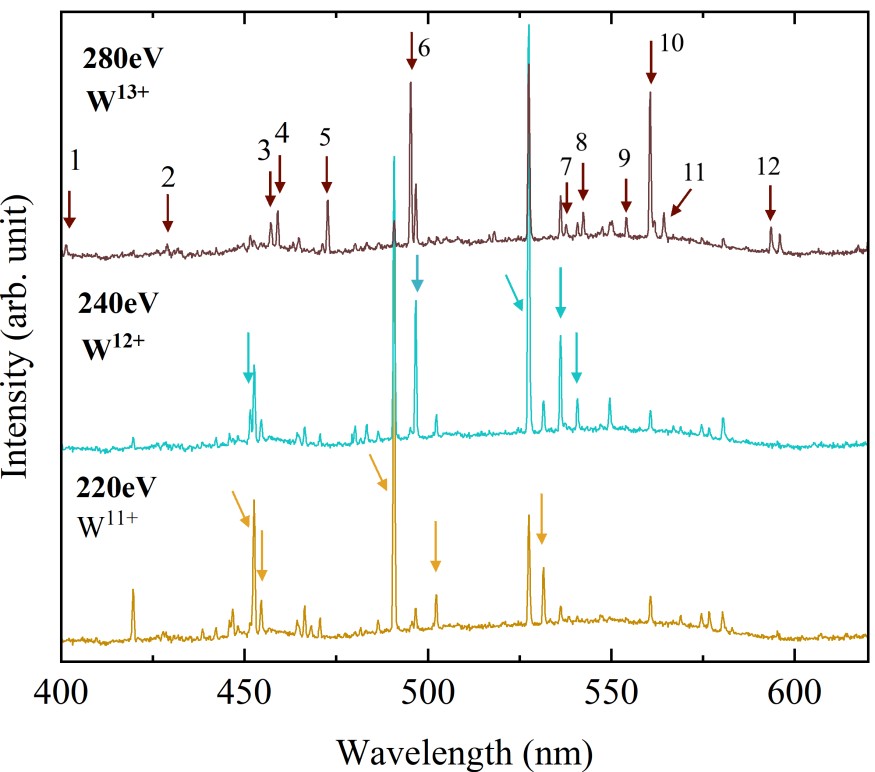

**Figure 1.** Experimental spectra observed at various electron energies. $W^{13+}$ lines are shown by arrows with numbers at 280 eV.

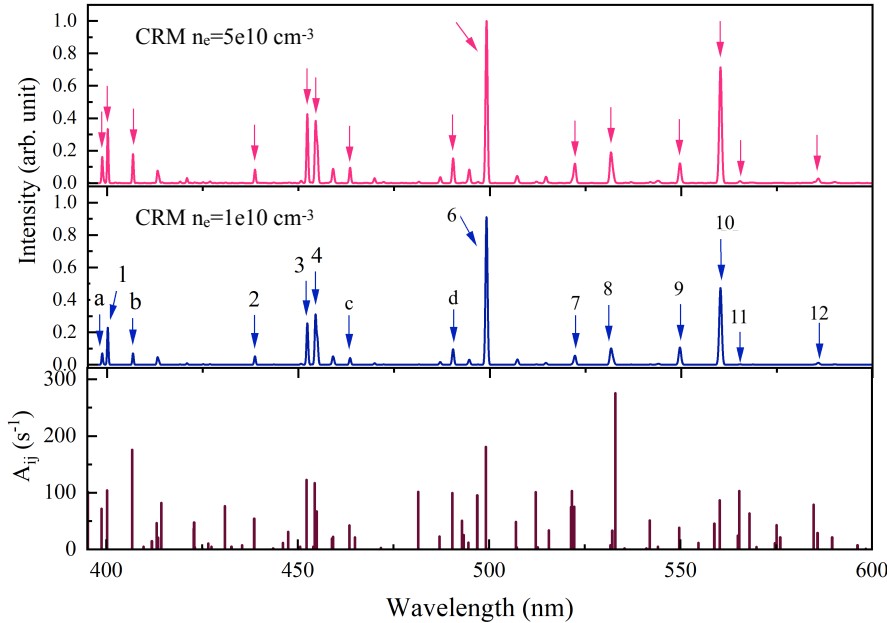

**Figure 2.** Theoretical spectra obtained at $E_e = 280$ eV and $n_e = 1 \times 10^{10}$ cm$^{-3}$ and $5 \times 10^{10}$ cm$^{-3}$. $W^{13+}$ lines, which were also observed in the experimental spectra at 280 eV, are shown by arrows with numbers and additional lines presented only in theoretical spectra are indicated by a, b, c, d. Lower panel shows the transition probabilities.

Wavelengths and transition probabilities of the most intense lines presented in the theoretical spectra are given in Table 1. The results obtained from the independent cal-

culation by GRASP2018 [11], along with the previously available results, are also given in Table 1. It can be seen that the GRASP calculations are closer to the experimental values. The maximum deviation from experimental and theoretical wavelengths calculated by FAC is 2.8%, while the agreement between experimental values and GRASP values is within 1.7%. Furthermore, the splitting of lower levels from the $4f^{13}$ configuration was consistent with the previous identifications by Kobayashi [9]. As shown in Table 1, the $4f^{13}$ splitting calculated in the present work using GRASP2018, were consistent with the value obtained by GRASP2K in [9]. Other prominent lines are identified as the M1 transition between $4f^{12}5p^1$ levels. One transition, near 542 nm, is identified as the transition between the lower levels of the $4f^{11}5p^2$ configuration. However, in the absence of GRASP calculations for $4f^{11}5p^2$ levels, the uncertainty could be large for this transition's wavelength. One intense line observed in the experimental spectra, around 472.68 nm, is absent in the present theoretical spectra. Recently, theoretical results for $W^{13+}$ were published by Liu et al. [10]. Both the theoretical results, presented in this work and [10], are closer to the experimental results reported by Kobayashi [9] than the measurements by Zhao et al. [7]. Our assignments for the two most intense lines, near 495.16 nm and 560.25 nm, and a line near 553.81 nm, are the same as obtained by [10]. However, the present identification of the 429.03 nm and 459.08 nm lines, is inconsistent with [10]. In fact, the transition corresponding to the 459.08 nm line in [10], is the same transition identified for the 457.26 nm line in the present work. In Liu et al.'s report [10], the 472.68 nm line is identified as the $((4\bar{f}^4)_4 5p_{1/2})_{J=9/2} \to ((4\bar{f}^5 4f^7)_5 5p_{1/2})_{J=9/2}$ transition, which corresponds to a wavelength of 559 nm in the present calculations. In addition, the Aij coefficient for this transition in the present calculation ($23\,\mathrm{s}^{-1}$) is similar to that reported in [10]. However, the theoretical intensity for this transition in both the calculations is much lower than the intensity observed in the experiment, therefore, we are not able to conclude that this transition actually belongs to the 472.68 nm line observed in the experimental spectra. In addition, a few new lines viz. 401.38 nm, 457.26 nm, 464.84 nm, 542.11 nm, and 593.28 nm, that were observed in the experimental spectra, are also identified in the present work, and our large-scale GRASP calculations support the present identifications. Furthermore, it is clear from Figure 2 that the line ratio 495.16 nm/560.25 nm of the most intense lines, has an electron density dependence, which can also be used as a diagnostic tool for several applications [19].

**Table 1.** Experimental and theoretical wavelengths (nm) and theoretical transition probabilities A ($s^{-1}$) for transitions in Pm-like $W^{13+}$.

| Label | $\lambda_{exp}$ (nm) | Upper Level | Lower Level | $\lambda_{FAC}$ (nm) | $\lambda_{GRASP}$ (nm) | $\lambda_{prev}$ (nm) | $A_{ijGRASP}$ ($s^{-1}$) | $A_{ijFAC}$ ($s^{-1}$) | $A_{ijprev}$ ($s^{-1}$) |
|---|---|---|---|---|---|---|---|---|---|
| a | | $((4\bar{f}^4)_4 5\bar{p}_{1/2})_{J=7/2}$ | $((4\bar{f}^5 4f^7)_3 5\bar{p}_{1/2})_{J=7/2}$ | 398.73 | 395.50 | 385.28 [b] | $7.22 \times 10^1$ | $7.39 \times 10^1$ | |
| 1 | 401.38 | $((4\bar{f}^5 4f^7)_4 5\bar{p}_{1/2})_{J=7/2}$ | $((4f^6)_4 5\bar{p}_{1/2})_{J=7/2}$ | 400.18 | 399.92 | 379.04 [b] | $1.06 \times 10^2$ | $1.03 \times 10^2$ | |
| b | | $((4\bar{f}^4)_2 5\bar{p}_{1/2})_{J=3/2}$ | $((4\bar{f}^5 4f^7)_2 5\bar{p}_{1/2})_{J=3/2}$ | 406.77 | 408.54 | 397.30 [b] | $1.80 \times 10^2$ | $1.76 \times 10^2$ | |
| 2 | 429.03 | $((4\bar{f}^5 4f^7)_2 5\bar{p}_{1/2})_{J=3/2}$ | $((4\bar{f}^5 4f^7)_3 5\bar{p}_{1/2})_{J=5/2}$ | 438.63 | 427.30 | 430.40 [a] 444.97 [b] | $4.27 \times 10^1$ | $4.69 \times 10^1$ | $7.42 \times 10^1$ [a] |
| 3 | 457.26 | $((4\bar{f}^4)_4 5\bar{p}_{1/2})_{J=9/2}$ | $((4\bar{f}^5 4f^7)_5 5\bar{p}_{1/2})_{J=11/2}$ | 452.31 | 454.18 | 457.76 [b] | $1.28 \times 10^2$ | $1.23 \times 10^2$ | |
| 4 | 459.08 | $((4\bar{f}^5 4f^7)_3 5\bar{p}_{1/2})_{J=5/2}$ | $((4f^6)_4 5\bar{p}_{1/2})_{J=7/2}$ | 454.45 | 457.13 | 458.90 [a] 442.46 [b] | $1.20 \times 10^2$ | $1.17 \times 10^2$ | $1.32 \times 10^2$ [a] |
| c | | $((4\bar{f}^4)_4 5\bar{p}_{1/2})_{J=7/2}$ | $((4\bar{f}^5 4f^7)_4 5\bar{p}_{1/2})_{J=9/2}$ | 463.48 | 463.61 | 455.95 [b] | $4.35 \times 10^1$ | $4.27 \times 10^1$ | |
| 5 | 472.68 | | | | | 473.80 [a] | | | $2.69 \times 10^1$ [a] |
| d | | $((4\bar{f}^5 4f^7)_2 5\bar{p}_{1/2})_{J=3/2}$ | $((4\bar{f}^5 4f^7)_3 5\bar{p}_{1/2})_{J=5/2}$ | 490.39 | 506.87 | 512.67 [b] | $1.00 \times 10^2$ | $9.97 \times 10^1$ | |
| 6 | 495.16 | $((4\bar{f}^5 4f^7)_5 5\bar{p}_{1/2})_{J=9/2}$ | $((4f^6)_6 5\bar{p}_{1/2})_{J=11/2}$ | 498.13 | 496.18 | 494.43 [a] 480.46 [b] | $1.86 \times 10^2$ | $1.81 \times 10^2$ | $1.80 \times 10^2$ [a] |
| 7 | 537.49 | $((4\bar{f}^4)_2 5\bar{p}_{1/2})_{J=5/2}$ | $((4f^6)_2 5\bar{p}_{1/2})_{J=5/2}$ | 522.23 | 528.35 | 521.37 [b] | $7.64 \times 10^1$ | $7.56 \times 10^1$ | |
| 8 | 542.11 | $((4\bar{f}^5 4f^6)5\bar{p}^2)_{J=13/2}$ | $((4\bar{f}^6 4f^5)5\bar{p}^2)_{J=15/2}$ | 534.60 | | 538.83 [b] | | $2.76 \times 10^2$ | |
| 9 | 553.81 | $((4\bar{f}^5 4f^7)_3 5\bar{p}_{1/2})_{J=5/2}$ | $((4f^6)_4 5\bar{p}_{1/2})_{J=7/2}$ | 549.65 | 552.30 | 552.43 [a] 535.12 [b] | $3.78 \times 10^1$ | $3.86 \times 10^1$ | $2.71 \times 10^1$ [a] |
| 10 | 560.25 | $(4f^7)_{J=7/2}$ | $(4\bar{f}^5)_{J=5/2}$ | 567.25 | 566.85 | 557.58 [a] 567.80 [c] | $8.16 \times 10^1$ | $8.70 \times 10^1$ | $8.86 \times 10^1$ [a] $8.39 \times 10^1$ [c] |
| 11 | 563.99 | $((4\bar{f}^5 4f^7)_5 5p_{3/2})_{J=9/2}$ | $((4f^6)_6 5p_{3/2})_{J=11/2}$ | 565.38 | 560.56 | 536.09 [b] | $1.05 \times 10^2$ | $1.04 \times 10^2$ | |
| 12 | 593.28 | $((4\bar{f}^5 4f^7)_1 5\bar{p}_{1/2})_{J=3/2}$ | $((4f^6)_2 5\bar{p}_{1/2})_{J=5/2}$ | 585.81 | 595.68 | 588.26 [b] | $2.90 \times 10^1$ | $2.92 \times 10^1$ | |

[a] by Liu et al. [10]; [b] transition energies are derived from the theoretical energy levels calculation by Singh et al. [6]; [c] results with GRASP2K [9].

## 4. Conclusions

To conclude, we have studied the visible spectra of $W^{13+}$ ions. Experimental spectra observed at 280 eV electron energy, between 400 and 600 nm, using CoBIT, were coupled with the C-R model to identify the respective lines. We also performed detailed RCI structure calculations using two individual codes, FAC and GRASP2018, to ensure the identifications. Most of the lines were identified as M1 lines, originating from intra-transitions between $5s^2 4f^{12} 5p^1$ levels. However, the line around 542 nm was from transition between the two lowest levels of $5s^2 4f^{11} 5p^2$, which needed to be verified from GRASP calculations. Further, more accurate theoretical works are still needed, to completely understand the electron correlations in this complex system.

**Author Contributions:** P. performed the initial investigation, formal analysis, and wrote the original draft. K.I. and M.F. performed the experiment. All the other authors helped with resources, writing, reviewing, and editing the manuscript. All authors have read and agreed to the published version of the manuscript.

**Funding:** This work was partially supported by the JSPS KAKENHI, grant number 18H01201, and NIFS Collaboration Research Program NIFS22KIIF018. Priti would like to acknowledge the financial support from the COE Fellowship, NIFS, Japan.

**Data Availability Statement:** Data will be made available on request.

**Conflicts of Interest:** The authors declare no conflict of interest.

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
