# Peer review of "Identification of Visible Lines in Pm-like W13+"

_atoms, doi:10.3390/atoms11030057_

Round 1

Reviewer 1 Report

The work is a worthwhile addition to the field and warrants publication. There are a small number of minor corrections/additions  that I recommend that the authors implement prior to the final publication of the manuscript.

Page 1: line 25, suggest that the order of the orbitals in the electronic configuration is changed: 4f^135s^2 -> 5s^2 4f^13 to be consistent with those over the page.

In relation to the formula on page 2, n+ has not been defined, this should be defined, I think  it is the number density from which RR transitions populate the jth level.

Additionally, is there any possibility of populating the jth level from ionisation from W12+?  This and RR from the W13+ jth level does  not seem  to  have been accounted for in the equation,  it would be helpful if the authors  explain why not.

Page 2: Line 63, English correction: "for are accounted in the model.", would be better as  "are accounted  for in the model."

Page 2: Line 77, Valanvce, should be Valance (x2 in VV).

Page 2: Line 78, "SD" does not appear to have been defined, it would help if it was.

Page 3: Figure 1. It would help to include the IP for  going  from W12+ to W13+ (~258eV)  somewhere in  the manuscript. Also the relatively strong line to the long wavelength side of 'line 7'  (280 eV spectrum) does not appear to be mentioned anywhere. If it is due to a lower  charge state, looks like it is in the 240eV spectrum  as well, this possibility should be mentioned (either in the caption  or the text) .

Page 3: Line 95. The authors  should either use C-R or CR for Collisional Radiative  not a mixture of the two  in the manuscript.

Page 4: Figure 2, The caption should include a  description of what is in the third panel (Aij).

Page 5: The standard of English  deteriorates significantly on this page of the manuscript. In particular there are many definite articles  (the) missing.

Also: Line 107, experiment should have a small e.

       Line 116, 'were published' rather than 'is published'. 

Line 122/123, Suggest 'Liu et al report' rather than 'In Liu et al report....'. Units are missing from 457.26.

Line 132 I would also suggest 'that the most intense line ratio 495.16nm/560.25nm has an electron density dependence ....',rather than the existing text. As it stands it is not clear whether the authors are referring to one  or more line ratios (I am assuming one).

Line 149 there is a uneccessary s.

Author Response

We want to thank all the referees for their valuable comments. We have carefully followed the reviewers’ comments and suggestions to revise our manuscript and have provided point-to-point responses below. We hope that the revised manuscript will be suitable for publication.

————————————————————————————————————

Referee Report #1

The work is a worthwhile addition to the field and warrants publication. There are a small number of minor corrections/additions that I recommend that the authors implement prior to the final publication of the manuscript.

Comment 1: Page 1: line 25, suggest that the order of the orbitals in the electronic configuration is changed: 4f^135s^2 -> 5s^2 4f^13 to be consistent with those over the page.

Response: We thank the reviewer for this suggestion, we have taken care of this point in the revised manuscript.

Comment 2: In relation to the formula on page 2, n+ has not been defined, this should be defined, I think it is the number density from which RR transitions populate the jth level. Additionally, is there any possibility of populating the jth level from ionization from W12+?  This and RR from the W13+ jth level does  not seem to have been accounted for in the equation,  it would be helpful if the authors explain why not.

Response: We have defined it in the revised manuscript on page number 2. In reference to the second comment, the simulation of W13+ spectra at 280 eV using a narrow gaussian of 5eV. Given that the ionization energy of W12+ is 258eV, it is expected that the density of W12+ ions will be minimal as most of the W12+ ions will have ionized to form W13+ ions. Consequently, we did not consider W12+ ions in the simulation of W13+ spectra at 280eV. Although considering multiple ionization states and the ionization balance including ionization and recombination processes is necessary for a more accurate intensity calculation, the current calculations are sufficient for line identification purposes. 

Comment 3: Page 2: Line 63, English correction: "for are accounted in the model.", would be better as  "are accounted  for in the model.” Page 2: Line 77, Valanvce, should be Valance (x2 in VV). Page 2: Line 78, "SD" does not appear to have been defined, it would help if it was.

Response: All of these errors have been rectified in the revised manuscript.

Comment 4: Page 3: Figure 1. It would help to include the IP for going from W12+ to W13+ (~258eV)  somewhere in the manuscript. Also the relatively strong line to the long wavelength side of 'line 7'  (280 eV spectrum) does not appear to be mentioned anywhere. If it is due to a lower charge state, looks like it is in the 240eV spectrum as well, this possibility should be mentioned (either in the caption or the text).                                                                                                     Page 3: Line 95. The authors should either use C-R or CR for Collisional Radiative not a mixture of the two in the manuscript.                                                                                                                 Page 4: Figure 2, The caption should include a  description of what is in the third panel (Aij).     Page 5: The standard of English deteriorates significantly on this page of the manuscript. In particular, there are many definite articles  (the) missing.                                                             Also : Line 107, experiment should have a small e.                                                                        Line 116, 'were published' rather than 'is published'.                                                                                Line 122/123, Suggest 'Liu et al report' rather than 'In Liu et al report....'. Units are missing from 457.26.

Response:  We have added the IP of W12+ ions in the revised manuscript. Also, the strong line suited near ‘line7’ belongs to W12+ (identified in previous experiments), and in the earlier draft we missed to indicate this in figure 1. Now we have added the arrow in figure 1 in the revised manuscript. All of the other suggestions related to minor errors have been taken into account. 

Comment 5: Line 132 I would also suggest 'that the most intense line ratio 495.16nm/560.25nm has an electron density dependence ....',rather than the existing text. As it stands it is not clear whether the authors are referring to one or more line ratios (I am assuming one). Line 149 there is a uneccessary s.

Response:  Yes, we are talking about a line ratio of 495.16nm/560.25nm lines only. The sentence has been modified to increase clarity in the revised manuscript.

Reviewer 2 Report

The manuscript presents experimental data on promethium-like tungsten W13+ ions. The experiment was performed at the compact electron beam ion trap (CoBIT) in Tokyo. Several optical lines of W13+ were measured and identified with the aid of collisional radiative model (CRM) which uses two sets of theoretical data from FAC and GRASP codes. The work contains important data for the fusion community and deserves publication. However, there are many issues, particularly in line identification, that need to be resolved before the paper can be accepted.

First of all, the paper lacks important details about the experimental setup and data analysis techniques. What type of optical spectrometer was used? What is the efficiency of the spectrometer? Importantly, how was the calibration of the spectrometer performed?

Second, details on the identification procedure are missing. The EBIT charge state distribution is somewhat broader, so how accurately were the W13+ lines identified. A look at top panel of Fig. 1 shows that some lines, shown as arrows and identified as W13+ lines, are visible even at lower electron beam energy. Could this be due to a metastable contribution? Is the electron beam energy shown in Fig. 1 corrected for space charge? Is there also a contribution from other impurity ions in CoBIT?

Third, I wonder why the authors used an electron temperature distribution in their CRM instead of a monoenergetic electron beam with a Gaussian distribution? I believe that the FAC/CRM module allows for the latter. Also, the authors mentioned in lines 131-133 that lines 7 and 11 are density sensitive. Why not use them to determine the realistic electron beam density in CoBIT and then use it in the CRM? This can definitely aid the identification since the modeled line intensity depends on density.

In the result section, the authors do not report the error in the determined wavelengths? What is the reason for this? From Tab. 1, I see that the difference between $\lambda_{exp}$ and $\lambda_{FAC}$ is both positive and negative, same between FAC and GRASP. So, the question is how confident are the authors in identifying the lines based on theoretical data and CRM. Did the authors check the difference between two consecutive lines in their experiment and in the theoretical data?

Author Response

Referee Report # 2

The manuscript presents experimental data on promethium-like tungsten W13+ ions. The experiment was performed at the compact electron beam ion trap (CoBIT) in Tokyo. Several optical lines of W13+ were measured and identified with the aid of collisional radiative model (CRM) which uses two sets of theoretical data from FAC and GRASP codes. The work contains important data for the fusion community and deserves publication. However, there are many issues, particularly in line identification, that need to be resolved before the paper can be accepted.

Comment 1:  First of all, the paper lacks important details about the experimental setup and data analysis techniques. What type of optical spectrometer was used? What is the efficiency of the spectrometer? Importantly, how was the calibration of the spectrometer performed?

Response: In the revised manuscript, now we have added the experimental details in brief. For detailed information, we have provided the references in the manuscript. 

Comment 2: Second, details on the identification procedure are missing. The EBIT charge state distribution is somewhat broader, so how accurately were the W13+ lines identified. A look at top panel of Fig. 1 shows that some lines, shown as arrows and identified as W13+ lines, are visible even at lower electron beam energy. Could this be due to a metastable contribution? Is the electron beam energy shown in Fig. 1 corrected for space charge? Is there also a contribution from other impurity ions in CoBIT?

Response:  Yes, spectra of higher charge states can appear at electron beam energy lower than the ionization energy of lower charge state. No, figure 1 is not corrected for space charge potential,  nevertheless, it is expected that the total shift is less than -10eV from our previous calculations. There were no possible contaminating elements that are heavier than tungsten. We have included the required details in the revised manuscript on page number 3.

Comment 3: Third, I wonder why the authors used an electron temperature distribution in their CRM instead of a monoenergetic electron beam with a Gaussian distribution? I believe that the FAC/CRM module allows for the latter. Also, the authors mentioned in lines 131-133 that lines 7 and 11 are density sensitive. Why not use them to determine the realistic electron beam density in CoBIT and then use it in the CRM? This can definitely aid the identification since the modeled line intensity depends on density.

Response: I'm sorry if it was unclear earlier. Just to clarify, we did mention in the previous manuscript that a Gaussian of 5 eV FWHM was used for rate calculations. We would like to emphasize that the determination of electron density is not straightforward, and therefore we refrained from attempting to extract information on the realistic electron density in this study. 

Comment 4: In the result section, the authors do not report the error in the determined wavelengths? What is the reason for this? From Tab. 1, I see that the difference between $\lambda_{exp}$ and $\lambda_{FAC}$ is both positive and negative, same between FAC and GRASP. So, the question is how confident are the authors in identifying the lines based on theoretical data and CRM. Did the authors check the difference between two consecutive lines in their experiment and in the theoretical data?

Response:  Now we have provided the uncertainty related to the measured wavelength (±0.05 nm) in the revised manuscript. Nevertheless, GRASP calculated wavelength for the transition between the same configuration has an almost negative shift.  In FAC calculation uncertainty is large. However, comparing the simulated spectra, energies as well as Aij coefficients from both (FAC and GRASP) calculations we can confirm the reliability of identification.

Reviewer 3 Report

This is a paper that follows a long traditions of spectroscopy of these elements, using EBIT's. The use of Grasp and FAC calculations to support identification is well established and the authors seems to mastering it. I think therefore this paper warrent publication, since it presents some new results for this particular system Pm-like W. It is short and concise, which is good for a well-establish technique that has been used by several groups (referred to in the paper).

I recommend publication after some support for minor English corrections.

Author Response

This is a paper that follows a long traditions of spectroscopy of these elements, using EBIT's. The use of Grasp and FAC calculations to support identification is well established and the authors seems to mastering it. I think therefore this paper warrants publication since it presents some new results for this particular system Pm-like W. It is short and concise, which is good for a well-establish technique that has been used by several groups (referred to in the paper).

I recommend publication after some support for minor English corrections.

Response: We appreciate your positive response. We made an effort to correct all the English errors.

Round 2

Reviewer 2 Report

After carefully reviewing the revised version of the paper, the authors' significant improvements to the presentation of their work are noteworthy. The authors' efforts in addressing all questions and incorporating the suggestions provided by the referees have resulted in a much clearer manuscript. As a result, the manuscript is recommended for publication with only minor issues.

While the authors calculated the rates using a Gaussian width of 5 eV in FAC CRM calculations, it is not entirely clear from lines 121-127 why temperature Te was used in Figure 2. It is unclear whether using a Maxwell-Boltzmann electron energy distribution would offer any advantages in identification process over using a Gaussian distribution.

Furthermore, determining density is not a straightforward process, as demonstrated in Nakamura et al, ApJ (2021), which the authors may cite accordingly. 

Lastly, it is noted that in line 151, the value should be 23 s^{-1}.

Author Response

We want to thank the referee for their valuable comments. We have carefully followed the reviewers’ comments and suggestions to revise our manuscript and have provided point-to-point responses below. We hope that the revised manuscript will be suitable for publication. 

Referee#2

After carefully reviewing the revised version of the paper, the authors' significant improvements to the presentation of their work are noteworthy. The authors' efforts in addressing all questions and incorporating the suggestions provided by the referees have resulted in a much clearer manuscript. As a result, the manuscript is recommended for publication with only minor issues.

Comment 1. While the authors calculated the rates using a Gaussian width of 5 eV in FAC CRM calculations, it is not entirely clear from lines 121-126 why temperature Te was used in Figure 2. It is unclear whether using a Maxwell-Boltzmann electron energy distribution would offer any advantages in the identification process over using a Gaussian distribution.

Response: Thank you for your suggestion. For a clearer picture, we have added the sentence “As an EBIT generates highly charged ions by successively ionizing trapped ions with a quasi-monoenergetic electron beam, theoretical spectra are obtained by including the rates of various processes calculated using narrow Gaussian electron energy distribution function with 5eV FWHM” near lines 124-126 in the revised manuscript. 

Comment 2: Furthermore, determining density is not a straightforward process, as demonstrated in Nakamura et al, ApJ (2021), which the authors may cite accordingly. 

Response: Thank you for your suggestions. We have added the reference [19] in the revised draft. 

Comment 3: Lastly, it is noted that in line 151, the value should be 23 s^{-1}.

Response: Thank you for pointing it out. We have corrected it in the revised manuscript. 

Round 3

Reviewer 2 Report

Based on the author's response to my comments and suggestions, I am satisfied that the necessary revisions have been made to clarify and improve the manuscript. Therefore, I recommend the manuscript as is for publication.